# QMILL: REPRESENTATIVE QUANTUM DATA GENERATION FOR QUANTUM MACHINE LEARNING UTILITY

## ABSTRACT

Quantum machine learning (QML) promises significant speedups, particularly when operating on quantum datasets. However, its progress is hindered by the scarcity of suitable training data. Existing synthetic data generation methods fall short in capturing essential entanglement properties, limiting their utility for QML. To address this, we introduce QMILL, a low-depth quantum data generation framework that produces entangled, high-quality samples emulating diverse classical and quantum distributions, thereby enabling more effective development and evaluation of QML models in representative data settings.

## 1 INTRODUCTION

Quantum machine learning (QML) is emerging as a transformative field, with applications ranging from image recognition to scientific computing (Riofrio et al., 2024; Liang et al., 2023; Peral-García et al., 2024; Wang et al., 2022; Guan et al., 2021). QML offers theoretical speedups over classical methods—but crucially, these speedups are provably attainable when operating on quantum datasets, i.e., data exhibiting superposition, interference, and entanglement (Biamonte et al., 2017; Carleo et al., 2019; DiBrita et al., 2024; Beaudoin et al., 2022; Hu et al., 2022; Delgado & Hamilton, 2022). Despite this, nearly all existing QML research focuses on classical data inputs due to the scarcity of real-world quantum datasets (Silver et al., 2022; 2023b). Quantum data is difficult to obtain: current quantum sensing technology is nascent, measurements are inherently probabilistic, large-scale data collection is cost-prohibitive, and noise from environmental and control sources further limits usability (Degen et al., 2017; Aslam et al., 2023). This data gap has become a fundamental bottleneck preventing the community from developing and validating QML models that can operate directly on quantum data, the very regime where QML promises a provable advantage.

Synthetic quantum data generation has therefore become critical to the future of QML. Without it, QML cannot meaningfully progress toward its theoretical potential, nor be ready when quantum-sensed data becomes more widely available in the coming years (Schatzki et al., 2021; Perrier et al., 2022). However, existing synthetic methods struggle to generate entanglement-rich datasets necessary for realistic QML workloads. One key metric is *concentratable entanglement* (CE), which captures inter-feature entanglement within a sample (Beckey et al., 2021; Schatzki et al., 2024; Liu et al., 2024; Jin et al., 2022). While Schatzki et al. (2021) introduced the first method to generate data with target CE values, their approach often fails to achieve the desired entanglement (deviations >20%), and assumes fixed CE across all samples—unlike real quantum datasets, which exhibit a natural distribution of CE values (Perrier et al., 2022; Medrano Sandonas et al., 2024).

To address these challenges, we present QMILL, a versatile quantum data generation framework designed to produce synthetic datasets that reflect diverse CE distributions and faithfully emulate both classical and quantum structures. The long-term role of synthetic quantum datasets remains an evolving question as quantum sensing and data-collection pipelines mature. QMILL is positioned as a pragmatic near-term tool that complements, rather than replaces, future real-world quantum data by enabling model development, benchmarking, and architecture-aware evaluation in the absence of large-scale quantum datasets. Our goal is to provide a practical foundation that enables QML research to progress as quantum data sources continue to evolve.

**This work makes the following key contributions:**

- QMILL generates synthetic datasets that capture a range of concentratable entanglement (CE) values, reflecting the variability observed in real-world quantum data.

- We design low-depth ansatzes tailored to Gaussian, Weibull, and Uniform distributions, enabling QMILL to stress-test statistical behavior under quantum constraints.

- By leveraging dual annealing (Sahin & Ciric, 1998), QMILL optimizes entangled states efficiently, ensuring compatibility with contemporary quantum hardware.

- QMILL incorporates SWAP tests (Zhang et al., 2024) to guarantee sample diversity and reduce redundancy, crucial for training generalizable QML models.

- We demonstrate QMILL's versatility across classical datasets (e.g., MNIST (Deng, 2012), FashionMNIST (Xiao et al., 2017), CIFAR-10 (Krizhevsky et al., 2009)) and quantum datasets (e.g., quantum chemistry (Perrier et al., 2022), soil moisture (Arumugam et al., 2024), dark matter (Chen et al., 2024)), achieving a deviation of $< 0.1$ from the target CE distributions.

- To show QMILL's practical utility, we train a quantum neural network on QMILL-generated CE feature sets and show an 84.8% accuracy against a classical ceiling.

- QMILL's data generation methodology, machine learning codebase, and generated datasets are open-sourced at: *https://anonymous.4open.science/r/QMill-FA93*.

## 2 BRIEF AND RELEVANT BACKGROUND

### 2.1 QUANTUM BITS, STATES, GATES, AND CIRCUITS

Quantum computing harnesses superposition and entanglement to unlock computational capabilities beyond classical systems (DiBrita et al., 2025; Ludmir et al., 2025). Its fundamental unit, the *qubit*, can exist in a superposition $|\psi\rangle = \alpha|0\rangle + \beta|1\rangle$, where $|\alpha|^2 + |\beta|^2 = 1$. This can be extended to an $n$-qubit system. Qubit systems reside in the $2^n$-dimensional Hilbert space, and quantum operations are performed using unitary gates. A sequence of gates forms a quantum circuit, which evolves an input state $|\psi_{\text{in}}\rangle$ to $|\psi_{\text{out}}\rangle = U|\psi_{\text{in}}\rangle$, where $U$ is the product of unitary gates.

### 2.2 VARIATIONAL QUANTUM CIRCUITS AND NOISE

Variational quantum circuits (VQCs), or *ansatzes*, are widely used in QML due to their tunability and expressiveness (Wang et al., 2022; DiBrita et al., 2024; Han et al., 2025). Each gate in a VQC is parameterized (e.g., $R_y(\theta)$), and the overall state $|\psi(\vec{\theta})\rangle = U(\vec{\theta})|\psi_0\rangle$ depends on a set of parameters $\vec{\theta}$. These parameters are optimized to minimize a classical loss function $f(\vec{\theta})$. On real hardware, especially NISQ devices, *gate noise* is a key challenge. As each gate has a non-zero error rate $\epsilon$, the total error grows with depth $d$ approximately as $1 - (1 - \epsilon)^d$ (Silver et al., 2023a; Bhattacharjee et al., 2019; Ash-Saki et al., 2019). Shallow circuits are therefore crucial to maintain high fidelity. QMILL leverages low-depth ansatzes to mitigate this noise while preserving expressivity.

### 2.3 QUANTUM DATASETS AND LIMITATIONS

Quantum data are most naturally represented as quantum states. An $n$-qubit datum is modeled by a density operator $\rho = \sum_{i,j=0}^{2^n-1} \rho_{ij}|i\rangle\langle j|$, where nonzero off-diagonal terms ($i \neq j$) encode entanglement. Algorithms such as quantum PCA, variational eigensolvers, and Hamiltonian learning can achieve exponential speedups when accessing such data directly from quantum memory (Lloyd et al., 2014; Wiebe et al., 2014). However, publicly available quantum datasets remain limited. Quantum chemistry datasets typically contain simple molecules like $H_2$, LiH, and $BeH_2$, yielding $\leq 6$ qubits after fermionic encoding (Perrier et al., 2022). Similarly, datasets from NV-center quantum sensors are restricted to a few qubits due to decoherence and control limitations (Qian et al., 2021; Zhang et al., 2023). Generating large-scale quantum datasets is both costly and experimentally challenging, which limits the scope of QML research.

### 2.4 CONCENTRATABLE ENTANGLEMENT (CE)

A critical challenge in synthetic quantum data generation is capturing realistic levels of entanglement. *Concentratable entanglement* (CE) quantifies the maximum entanglement that can be localized between subsystems of a quantum state (Beckey et al., 2021; Schatzki et al., 2024; Liu et al.,

2024; Jin et al., 2022). For a bipartite split $\{A, B\}$ of a state $\rho$, CE is defined as:

$$C_E(\rho) = \max_{\rho_{AB}} S(\text{Tr}_B(\rho_{AB})),$$

where $S(\rho) = -\text{Tr}(\rho \log \rho)$ is the von Neumann entropy. Beckey et al. (2021) provide an efficient method for computing CE in many relevant cases. CE serves as a proxy for "quantumness" in data. High CE enables QML models to leverage entanglement for improved performance, particularly in domains such as quantum chemistry (Perrier et al., 2022).

# 3 MOTIVATION FOR QMILL

Progress in QML is hindered by the scarcity of scalable, diverse, and entanglement-aware quantum datasets. Existing quantum datasets are small and expensive to generate, and current synthetic methods are even more limited (Zoufal et al., 2019; Benedetti et al., 2019). The most notable effort by Schatzki et al. (2021) proposes training ansatzes to match a fixed CE t; however, their approach often fails to reach the desired CE value and overlooks a more fundamental issue: real quantum data does not have a single entanglement level. In practice, quantum datasets exhibit a spread of CE values across samples. Training and benchmarking QML models on a fixed CE setting oversimplifies the problem and leads to poor generalization.

While CE is not the only meaningful descriptor of quantum correlations, we focus on it because it provides a tractable, hardware-efficient summary statistic that still preserves sample-level variability. CE also offers an interpretable proxy for global multi-qubit structure that many QML models rely on, without requiring full state tomography or cost-prohibitive estimators. Our goal is not to treat CE as a complete or sufficient characterization, but to show that matching its distribution represents a necessary step beyond prior work that targets a single entanglement value and thereby collapses intra-dataset structure. What is needed instead is a generator that can produce datasets with controlled CE distributions, capturing the full range from weak to strong entanglement. QMILL *fills this gap. It generates synthetic datasets where CE values follow user-specified distributions. It uses low-depth, distribution-specific ansatzes optimized via annealing methods, making it both noise-resilient and efficient. The result is a scalable framework for producing entanglement-rich, diverse, and realistic quantum datasets, enabling the next stage of data-driven QML development.*

# 4 QMILL'S DESIGN

QMILL is a quantum data generation framework designed to create high-quality synthetic datasets for QML. Its core goal is to generate entangled states that match a target distribution of concentratable entanglement (CE) while remaining shallow enough to run on noisy hardware. As shown in Fig. 1, QMILL starts with Haar-random

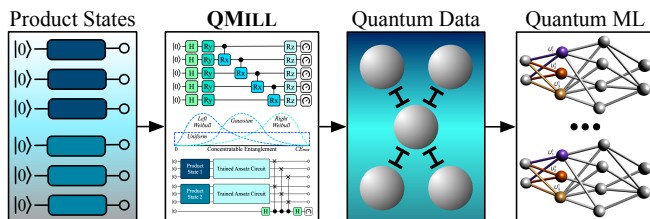

Figure 1: QMILL takes classical product states and generates diverse and customizable quantum data for QML tasks.

product states, applies a parameterized ansatz to entangle them, optimizes the circuit to match a CE distribution, and validates sample diversity in the generated dataset via SWAP tests.

The framework has four components: (A) a set of low-depth variational ansatzes supporting different entanglement structures, (B) a pipeline for sample generation and CE measurement using efficient density matrix approximations, (C) a dual-annealing optimization loop minimizing CE distribution mismatch, and (D) a SWAP test–based diversity check to avoid mode collapse. Together, these components make QMILL scalable, customizable, and hardware-compatible.

## 4.1 PARAMETERIZED CIRCUITS & OBJECTIVE FUNCTION

The primary design tension lies between expressibility and hardware feasibility: deeper circuits can model richer CE distributions, but are more susceptible to noise on near-term hardware. To explore this trade-off, QMILL includes four low-depth parameterized circuits (A1–A4), shown in Fig. 2, each probing different entanglement and noise behaviors. A1 uses compact RX, RZ, and

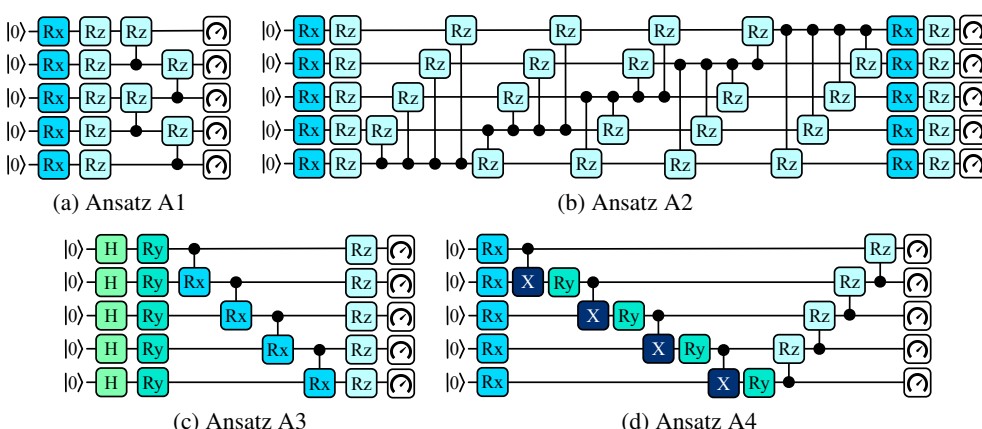

(a) Ansatz A1                 (b) Ansatz A2

(c) Ansatz A3                 (d) Ansatz A4

Figure 2: QMILL develops a variety of ansatz designs for real and synthetic CE distributions.

controlled-RZ gates. A2 extends A1 with a denser entangling pattern. A3 incorporates Hadamard and controlled-RX gates. A4 combines RX, RY, CNOT, and controlled-RZ gates in a deeper structure. We note that optimizing an ansatz is a challenging and well-studied problem space, and is orthogonal to the goals of this work. Rather than performing an expansive or costly architectural search, we intentionally restrict our study to hardware-efficient ansatz families motivated by prior work (e.g., (Sim et al., 2019)) and commonly considered in NISQ-era deployments. The variability observed across architectures (explored in Sec. 6) provides empirical guidance on which lightweight ansatz structures tend to align best with different CE distribution shapes, enabling practical and distribution-aware selection without the need for extensive tuning.

The goal is not to find a universal best ansatz, but to evaluate which structures best match the target CE under depth constraints. Parameters $\vec{\theta}$ are tuned using dual annealing (Sahin & Ciric, 1998), a global optimizer effective in non-convex landscapes where gradient methods often fail, especially for skewed or multimodal CE targets. The objective is to minimize the total variation distance (TVD) between the empirical CE histogram and the target:

$$C(\vec{\theta}) = \text{TVD}(P_{\text{generated}}(\vec{\theta}), P_{\text{target}}), \text{TVD}(P, Q) = \tfrac{1}{2} \sum_x |P(x) - Q(x)|.$$

TVD provides a symmetric, distribution-agnostic penalty, making it well-suited for our task. Note: we compute TVD over one-dimensional CE histograms rather than full output state distributions, which keeps the cost polynomial in the number of generated samples rather than exponential in qubit count. Since CE is estimated through measurement-based surrogate quantities rather than full state tomography, TVD avoids scaling challenges while still capturing distributional structure relevant for QML evaluation.

## 4.2 SAMPLE GENERATION AND CE MEASUREMENT

Sample generation begins with product states drawn from the Haar measure:

$$|\psi\rangle = \cos(\theta/2)|0\rangle + e^{i\phi}\sin(\theta/2)|1\rangle, \theta \sim U(0, \pi), \ \phi \sim U(0, 2\pi).$$

These unentangled inputs allow clear control over the entanglement introduced by the circuit.

To measure CE, QMILL considers an efficient approximation from Beckey et al. (2021):

$$\text{CE}(\rho) = 1 - \tfrac{1}{2^{c(s)}} \sum_{\alpha \in \mathcal{P}(s)} \text{Tr}[\rho_\alpha^2],$$

where $\rho_\alpha$ is the reduced density matrix over subset $\alpha$, and $\mathcal{P}(s)$ is the power set of all qubit subsets. This method captures entanglement via subsystem purities and enables CE estimation without tomography. However, its measurement cost scales with $|\mathcal{P}(s)|$, which becomes impractical beyond small $n$. Thus, we employ estimators that preserve the ordering signal required for model selection, and utilize linear shot budgets (see Appendix A.1 for details on definitions, bounds, and scalability).

## 4.3 CONCENTRATABLE ENTANGLEMENT DISTRIBUTIONS

A core strength of QMILL is its ability to match full distributions of CE values, not just a single entanglement target. This is essential because real quantum datasets rarely have uniform entanglement; instead, they exhibit broad or skewed CE profiles. Supporting full CE

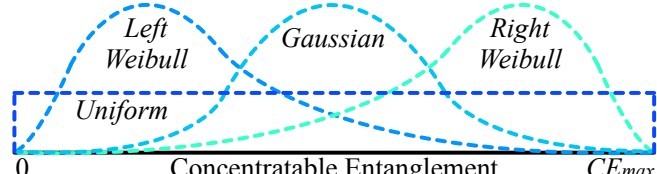

Figure 3: In addition to the CE distributions of real data, QMILL also tests its efficacy for different CE distributions.

distributions enables realistic benchmarking of QML models across diverse entanglement regimes. QMILL supports both real and synthetic targets. For real CE distributions, we extract histograms from quantum-encoded classical datasets such as MNIST, FashionMNIST, and CIFAR-10 (Krizhevsky et al., 2009; Xiao et al., 2017; Deng, 2012), as well as native quantum datasets like quantum chemistry, soil moisture, and dark matter (Arumugam et al., 2024; Chen et al., 2024; Schütt et al., 2017). Each dataset is amplitude encoded, and CE is computed to produce empirical histograms used as generation targets. To stress-test QMILL's flexibility, we define several synthetic CE distributions:

- **Uniform:** Entanglement spread evenly from 0 to $CE_{\max}$.
- **Gaussian:** Most samples cluster around moderate CE.
- **Weibull (Left/Right):** Skewed distributions representing mostly low or high entanglement.

Fig. 3 shows target examples. During training, QMILL bins CE values from generated samples and compares them to the target via TVD. This approach enables the controlled exploration of how QML models respond to different entanglement regimes. For instance, one can test how ansatz performance varies under low versus high CE, or compare the demands of classical and quantum datasets. QMILL *thus enables entanglement-aware dataset engineering, which comprises more than just data generation.*

### 4.4 SWAP TEST FOR SAMPLE DIVERSITY VALIDATION

Matching CE distributions alone doesn't guarantee dataset quality. A generator could produce near-identical states with the same CE, resulting in low diversity and poor generalization. Ensuring that QMILL outputs not only entangled but also distinct samples is therefore critical. To enforce diversity, QMILL uses the SWAP test (Zhang et al., 2024) (see Appendix A.2 for details), a quantum routine that measures the fidelity between two states:

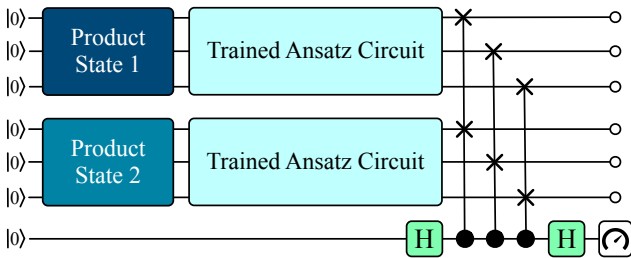

Figure 4: QMILL uses the SWAP test to validate the dissimilarity of any two random samples with similar CE values.

$$P(|0\rangle) = \tfrac{1}{2} \cdot (1 + |\langle\psi|\phi\rangle|^2)$$

High fidelity ($\approx 1$) indicates similarity; values near $0.5$ suggest dissimilarity. Unlike classical similarity checks, the SWAP test is efficient and non-destructive. During training, a random subset of sample pairs is selected, and their average SWAP test score is calculated. If average fidelity exceeds a threshold (e.g., $> 0.95$), this signals mode collapse. In response, QMILL introduces a diversity penalty to steer optimization away from redundant states, especially important for sharp or skewed CE targets. The SWAP test is practical because it requires only up to three-qubit controlled operations, which decompose into standard one- and two-qubit gates on hardware without native multi-qubit support. Each test uses $2n+1$ qubits with shallow depth, leading to linear growth in qubit count rather than exponential growth in depth. This is compatible with NISQ hardware, which typically tolerates larger qubit footprints more easily than deep circuits due to high qubit decoherence noise. By combining CE alignment with active diversity enforcement, QMILL *produces datasets that are both representative of the target entanglement structure and richly varied at the state level, generating diverse dataset samples.*

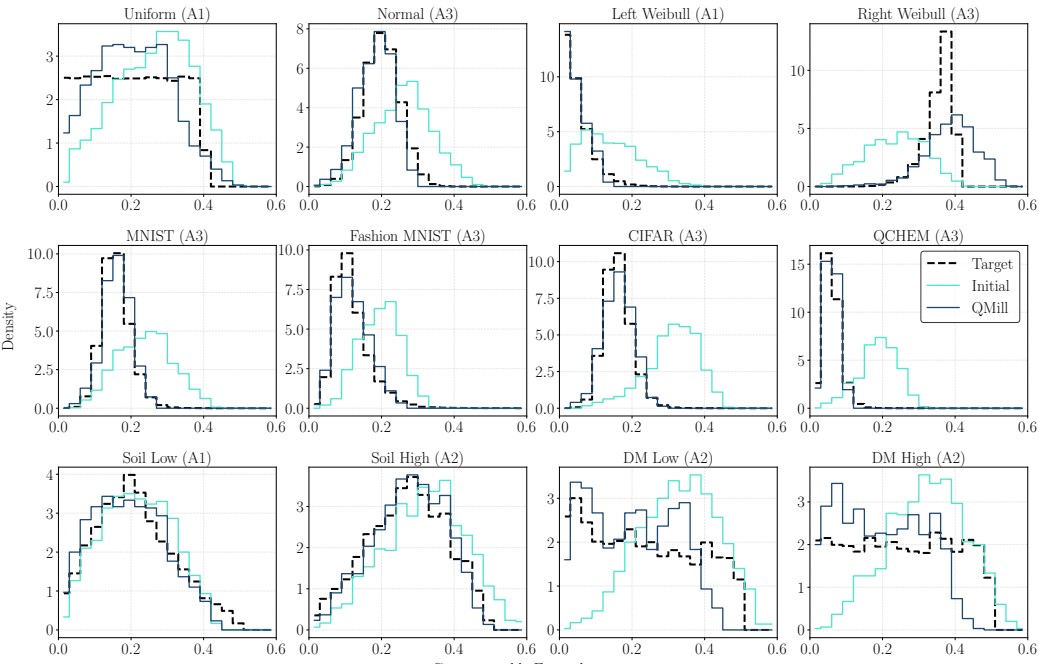

Figure 5: Showcase of top-performing circuits training to mimic the CE of various arbitrary, stress-testing, and real-dataset distributions.

**In summary,** QMILL unifies low-depth ansatzes, CE-targeted optimization, and diversity validation into a practical pipeline for generating high-quality quantum datasets. Each component addresses a key challenge, circuit noise, CE fidelity, and sample uniqueness, resulting in a scalable framework ready for QML training and evaluation. Finally, because QMILL relies on shallow, hardware-efficient circuits, its design is portable across backends and does not assume device-specific gate libraries or calibration profiles. To empirically validate this, we performed noisy simulations on a neutral atom architecture using the Bloqade SDK. We successfully transpiled our optimized ansatz to the native gate set via global Rydberg pulses and executed it under the GeminiOneZoneNoise-Model. The simulation yielded a Concentratable Entanglement value of 0.3175 with a global state purity of 0.68. This performance is within the same order of magnitude as results observed on super-conducting backends approximately 0.46, confirming that QMILL's low-depth ansatzes effectively generalize across varying noise environments and hardware implementations.

## 5 QMILL'S IMPLEMENTATION AND METHODS

### 5.1 EXPERIMENTAL AND SOFTWARE SETUP

We evaluate QMILL using Qiskit Aer's circuit simulator with IBM Sherbrooke's noise model for noisy simulations. Real-machine experiments are also conducted on IBM Sherbrooke. All circuits are implemented in Python 3.10.12 using Qiskit 1.2 (Aleksandrowicz et al., 2019). Simulations are executed on a local research cluster running Ubuntu 22.04.2 LTS, with a 32-core 2.0 GHz AMD EPYC 7551P processor and 32 GB RAM. Each experiment uses 2048 measurement shots. Circuits are constructed using Qiskit's `QuantumCircuit` class, and noiseless simulations are performed for baseline evaluations. Empirically, we observe that optimization time scales approximately linearly with the number of circuit parameters for the low-depth ansatz families considered, and linearly with the number of measurement shots, consistent with our expectation.

### 5.2 EVALUATED CLASSICAL AND QUANTUM DATASETS

To evaluate QMILL's ability to generate quantum data with controlled CE characteristics, we use both synthetic and real datasets. For stress-testing, we define four synthetic CE target distributions over the interval [0, 0.4]. These include a uniform distribution, a Gaussian distribution centered

at 0.2 with a standard deviation of 0.05, a left-skewed Weibull distribution (shape parameter 1.2, scaled by 0.05), and a right-skewed variant obtained by reflecting the left-skewed version across $x = 0.2$. These distributions are chosen to span a wide range of entanglement behaviors observed in real quantum systems. We also evaluate CE profiles derived from classical datasets: MNIST (Deng, 2012), FashionMNIST (Xiao et al., 2017), and CIFAR-10 (Krizhevsky et al., 2009).

The data are standardized and reduced in dimension using Principal Component Analysis (PCA) (Pearson, 1901) to $2^n - 1$ features for $n$ qubits. PCA is applied solely to reduce the dimensionality of classical datasets so they can be amplitude encoded using a small and hardware-compatible qubit count. Direct amplitude encoding of full-feature vectors would require substantially more qubits than can be simulated today, but this is a limitation of available hardware and classical simulation resources rather than of QMILL, which does not inherently depend on PCA or small state sizes. These features are then embedded into quantum amplitudes using amplitude encoding (Rath & Date, 2024). The resulting quantum states are processed to compute CE values as described in Sec. 4, and their empirical CE distributions are scaled for comparison against QMILL outputs. More significantly, we evaluate CE targets extracted from three quantum datasets. The quantum chemistry dataset (Perrier et al., 2022) contains 134k molecules from QM9, each represented using engineered features derived from atomic and molecular statistics. These include atomic charge moments, vibrational frequencies, spatial metrics, and element counts, all of which are aggregated into fixed-length vectors suitable for encoding.

For quantum datasets, we consider two protocols. The first is a soil moisture sensing setup based on the STQS framework (Jebraeilli et al., 2025; Arumugam et al., 2024), which utilizes entangled Rydberg atoms to detect phase differences from soil reflections. Simulations are run for both high- and low-moisture regimes, incorporating phase jitter to generate ensembles of quantum states. CE values are computed for each state to form CE distributions reflective of different sensing environments. The second protocol is a dark matter detection setup adapted from (Chen et al., 2024), using a four-qubit sensing circuit where the signal strength $\phi$ encodes the dark matter interaction. Simulations with $\phi = 0.01$ and $\phi = 0.1$ yield distinct CE distributions, enabling us to evaluate QMILL under both weak and strong signal conditions. See Appendix B for sensor circuit details.

We use circuits with 3–5 qubits, depending on the number of features to be generated for an application. SWAP-based validation requires $2n + 1$ qubits (e.g., 21 qubits for a 10-qubit circuit), which is costly to simulate without HPC resources, and near-term limitations in error correction necessitate simulation for controlled evaluation. Our choice of smaller circuit sizes is therefore driven by current practical constraints rather than by a technical limitation of the approach.

### 5.3 QMILL'S EVALUATION METRICS

We evaluate the ansatz performance using four key metrics. The **TVD** measures how well the ansatz can reproduce target CE distributions, with lower values indicating better performance. The **TVD variance** quantifies the consistency of the ansatz across different distributions, where lower variance suggests more reliable performance. We also compute the **TVD rank** by comparing the TVD of each ansatz against those of others for all distributions, assigning ranks 1 through 4 to each distribution (with 1 being the best performing), and then averaging these ranks across all distributions.

We use the **SWAP test similarity** to compare two quantum states by measuring their similarity, yielding a probability $P(|0\rangle)$ between 0.5 (distinct states) and 1.0 (identical states). For statistical robustness, we perform multiple SWAP tests within each CE range, with the number of tests limited by the available states in that range. For each circuit architecture and target distribution, we first generate 1000 random product states and transform them through the trained ansatz. The resulting states are then grouped by their CE values into discrete ranges. Within each range, we randomly pair states and perform SWAP tests between them.

## 6 QMILL'S EVALUATION AND ANALYSIS

### 6.1 QMILL'S ABILITY TO CAPTURE DISTRIBUTIONS

We evaluate QMILL across multiple CE distributions, observing varied performance depending on the target shape. Fig. 5 presents the best-performing ansatz for each case. We note that as prior

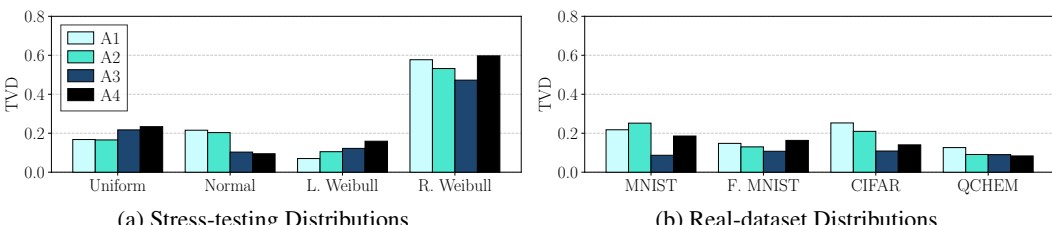

(a) Stress-testing Distributions  (b) Real-dataset Distributions

Figure 6: QMILL's TVD performance on (a) arbitrary distributions used for stress testing its impact and (b) real datasets (lower is better).

work does not target CE distributions, there is no state-of-the-art method for this area. Fixed-target methods can be used as a reference point, but they are fundamentally optimized for a different objective. We therefore compare QMILL with our baselines that reflect competitive alternatives under the same distribution-matching goal.

For the uniform distribution, ansatz A1 achieves a TVD of $\approx 0.18$, reflecting reasonable spread coverage. In the Gaussian case, ansatz A4 performs best, achieving a TVD below $0.1$ (Fig. 6(a)). The trained distribution accurately captures both the central peak and the bell-shaped spread, closely matching the target. Ansatz A1 also performs well on the left-skewed Weibull, effectively modeling the sharp peak and gradual decline. The right-skewed Weibull, however, proves more challenging: although ansatz A3 reduces the TVD to 0.5, the improvement over the initial state is modest. This distribution intentionally concentrates CE at unrealistically high values to stress test QMILL's limits. We emphasize that the high-CE right-skewed target is intentionally unrealistic: achieving arbitrarily high entanglement from arbitrary input product states is fundamentally constrained by quantum reversibility. A circuit that could reliably map many distinct inputs to a fixed, highly entangled output would be invertible and could therefore be used to generate arbitrary states from that output, which is not physically consistent. The right-skewed case is therefore included as a stress test rather than an achievable target. *Despite these extremes,* QMILL *achieves reasonably low TVD across all cases, demonstrating robustness even under adversarial conditions.*

## 6.2 QMILL'S ABILITY TO EMULATE REAL DATASETS

QMILL shows strong performance when emulating CE distributions from real-world classical and quantum datasets. Across all evaluated datasets, the trained distributions align closely with targets, with high-fidelity matches observed in most cases. For MNIST, ansatz A3 achieves a TVD $< 0.1$, significantly outperforming A1 and A2 and accurately reproducing the characteristic bell-shaped CE profile (Fig. 5, Fig. 6(b)). Similar performance is observed for FashionMNIST and CIFAR, with QMILL consistently narrowing the initial CE spread to better match the target structure.

On quantum datasets, QMILL performs especially well. For the quantum chemistry dataset, all ansatzes yield TVD values below 0.2, despite the narrow CE band, and the results for the soil moisture and dark matter datasets similarly show close alignment (Fig. 5, Fig. 7). While later evaluations show some ansatzes outperform others overall, these results highlight that

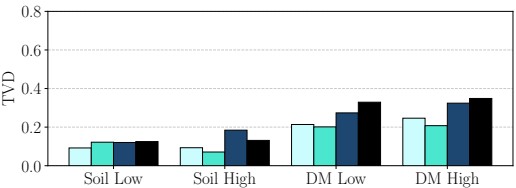

Figure 7: QMILL's TVD with quantum sensors.

different architectures excel on specific distributions. For example, A3 is best suited for MNIST, A2 performs well on soil and DM sensor signals, A1 is optimal for the Left Weibull dataset, and A4 captures the chemistry dataset most effectively. This underscores the utility of maintaining a diverse ansatz library tailored to different CE profiles.

## 6.3 DIVERSITY OF SAMPLES GENERATED BY QMILL

We assess the diversity of generated states using SWAP tests between state pairs within similar CE ranges across all four circuit architectures. As shown in Fig. 8(a), each point represents the average SWAP test value for a given CE range, with point size indicating the number of state pairs tested (the larger the circle, the more the samples). Most values lie between 0.5 and 0.6, suggesting that generated states are largely distinct, even within the same CE bin. We observe slightly higher similarity in

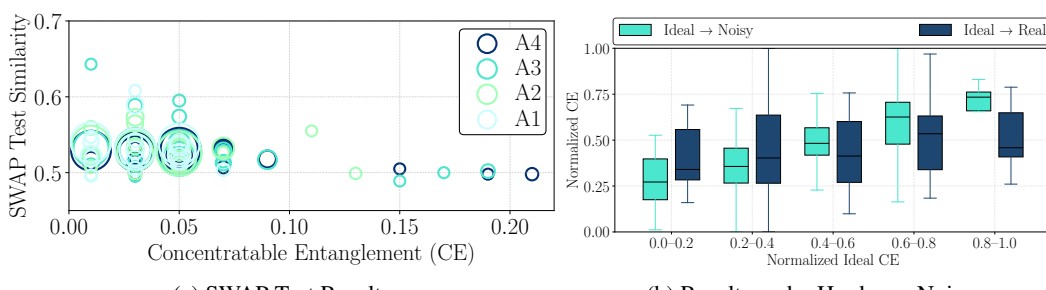

(a) SWAP Test Results        (b) Results under Hardware Noise

Figure 8: (a) SWAP test results across different CE values. Each point represents a set of SWAP tests between states with similar CE values. The y-axis shows the test outcome (0.5 indicates distinct states, while 1.0 indicates identical states), and the x-axis shows the CE value of the tested states. (b) CE differences between ideal simulation, noisy simulation, and real hardware for the soil moisture dataset highlight the performance differences under different scenarios.

the low CE regime, where most samples are concentrated—an expected outcome, as high-CE states are harder to generate. In contrast, states in higher CE ranges consistently yield SWAP scores near 0.5, indicating strong sample-level diversity. This trend is consistent across all ansatzes, confirming that QMILL reliably produces non-redundant states across the full CE spectrum.

## 6.4 QMILL'S PERFORMANCE UNDER NOISE

To evaluate robustness under realistic conditions, we compare CE values for quantum states from the soil moisture dataset across ideal simulation, noisy simulation (using IBM Sherbrooke's noise model), and real hardware execution on IBM Sherbrooke (Fig. 8(b)). Interestingly, both noisy simulation and real hardware runs exhibit higher CE values than ideal simulation, likely due to noise-induced deviations reducing the likelihood of measuring the all-0 state. While all three settings capture a similar trend (approximately linear), real hardware consistently shows more variance than noisy simulation. This suggests that IBM's noise model slightly underestimates noise effects compared to actual device behavior. These results emphasize the need to evaluate QML-relevant quantum datasets under both simulated and real hardware conditions, as noise can significantly influence measured entanglement.

## 6.5 PERFORMANCE OF DIFFERENT ANSATZES

We compare the four ansatz designs using mean TVD, median TVD, TVD variance, and average rank across all target CE distributions (Fig. 9). Ansatz A3 delivers the best overall performance, achieving the lowest mean and median TVD along with low

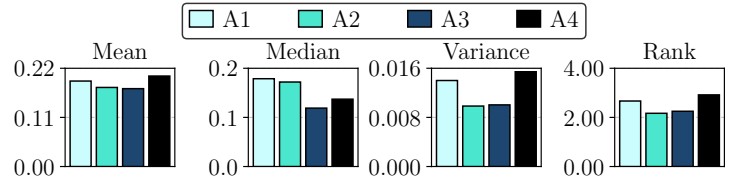

Figure 9: The aggregated TVD performance of the different ansatzes shows that A3 performs the best in general.

variance. Despite its simplicity, featuring only a single layer of controlled operations and Hadamard-based state preparation, A3 strikes an effective balance between expressivity and depth. A2 performs comparably in terms of accuracy and rank, but has significantly higher depth due to its extensive use of controlled-RZ gates, offering no clear performance gain over A3. A1, the simplest in structure, shows the weakest results, with the highest median TVD, indicating that minimal circuits lack sufficient expressivity to model CE distributions effectively. A4 offers balanced performance with moderate depth; its use of X gates provides a slight improvement over rotation-only designs, but still falls short of A3's efficiency. The results suggest that while simplicity helps with noise resilience, a minimal level of entangling structure is essential. A3 best captures this trade-off.

## 6.6 DEPLOYING QMILL FOR QML CLASSIFIERS

We now demonstrate the practical utility of QMILL, specifically whether these synthetic CE datasets can effectively train QML models. We train a three-qubit QNN on QMILL-generated CE feature

Table 1: Performance (%) of dual-annealing-optimized QNNs comparing ideal vs. noisy circuits.

| Accuracy | Precision | Recall | F1 Score |
|---|---|---|---|
| Ideal: 81.8, Noisy: 84.8 | Ideal: 83.3, Noisy: 87.0 | Ideal: 83.3, Noisy: 84.5 | Ideal: 83.2, Noisy: 83.8 |

sets under both ideal and noisy simulations, and benchmark its accuracy against a classical logistic-regression ceiling. We first create a dataset using the quantum soil sensor data we generated earlier by batching the CE values into 400 samples, with each sample containing 9 CE values, and then assigning the label 0 or 1 depending on whether the samples came from low- or high-moisture soil. We implement a three-qubit classifier by encoding each of the input CE value features into an RY–RX–RZ feature map, then applying an ansatz with full entanglement between qubits using Qiskit's RealAmplitudes parameterized circuit. The model then measures a single-qubit Z observable on the first qubit and feeds the expectation value into a QNN.

Training is performed using a dual-annealing optimizer, and performance is evaluated through 5-fold cross-validation. Noise is modeled using IBM Sherbrooke's error parameters. Table 1 summarizes accuracy, precision, recall, and $F_1$ score for the noisy and ideal circuits, each normalized against a classical logistic-regression baseline set to 100%. Notably, the noisy implementation falls within a few percentage points of its ideal counterpart, demonstrating that our three-qubit classifier retains nearly all of its predictive power even in the presence of realistic gate and readout errors. The slightly higher accuracy observed under the noisy setting in can be attributed to sampling variance rather than a systematic performance gain from noise, and conclude that the two settings exhibit comparable accuracy in practice. *This tight correspondence confirms that, for Concentratable Entanglement–based features, the modest noise levels expected on near-term quantum hardware will not adversely affect* QMILL's *performance for QML applications.*

## 7 RELATED WORK

As QMILL is the first-of-its-kind effort toward synthetic QML data generation, the related work is limited. Schatzki et al. (2021) attempted to generate entangled datasets using quantum circuits trained to achieve a single target, concentratable entanglement value. However, this approach falls short as generated samples often deviate from the desired entanglement. Xu et al. (2025) employed supervised QML and CE lower bound metric to generate mixed-state datasets designed for entanglement classification around a target value, which is orthogonal to our approach of generating target CE distribution datasets. Zhang et al. (2025) uses a denoising model to synthesize class-specific GHZ/W-like states; unlike QMILL, this does not control CE distributions across datasets nor enforce sample diversity.

Other approaches include domain-specific methods, such as Quantum Generative Adversarial Networks (QGANs) for detecting product states (Steck & Behrman, 2024), and quantum transfer learning on small, high-dimensional datasets for remote sensing (Otgonbaatar et al., 2023). While innovative, these methods do not generalize to QML tasks requiring flexible entanglement distributions. (Yu et al., 2023) proposed generating optimal datasets for learning unitary transformations, yet the approach remains constrained to classical applications. Sim et al. (2019) explored the expressibility of parameterized quantum circuits, providing insight into ansatz selection, but in our work, we observe that higher expressibility does not necessarily correlate with better CE matching. This limitation necessitated the design of a customized ansatz in QMILL to better align with targeted CE distributions, enabling more effective synthetic data generation across a range of entanglements.

## 8 CONCLUSION

We introduced QMILL, a quantum data generation framework that produces diverse datasets with distributions of concentratable entanglement values, supporting robust QML model development. By leveraging customizable ansatz and efficient, low-depth circuits with SWAP tests, QMILL enables scalable, high-quality synthetic data generation with a diverse set of samples, validated across multiple classical and quantum datasets. QMILL thus addresses a critical need in QML, providing an essential framework for quantum data generation that advances QML training and evaluation, ultimately enabling quantum utility and speedup in practice.

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

# A    DESCRIPTION OF METRICS

## A.1    CONCENTRATABLE ENTANGLEMENT (CE)

**Motivation and intuition.**    Concentratable Entanglement is a measure of multipartite entanglement. Intuitively, a quantum state with high CE has its entanglement broadly distributed across many different partitions of qubits, indicating a complex, global correlation structure. In contrast, a low-CE state may have its entanglement localized to a small number of qubits. For QML, high-CE states are of interest as they provide a highly correlated structure that quantum algorithms can exploit for a potential advantage. The formal definition of CE is based on the average **purity** of all subsystems of a given size, where purity is a measure of how much we know about a quantum state and is a value between 0 and 1 that tells us whether a state is pure or mixed: 1 means perfectly known pure state, lower values mean it is noisy or mixed. For a piece of a larger, globally pure system, any drop in purity is because that piece is entangled with the rest. CE averages these purities over many pieces, so a lower average purity means the entanglement is more widely spread.

Formally, the purity of a quantum state for a subsystem $S$, described by the density matrix $\rho_S$, is given by $\mathrm{Tr}(\rho_S^2)$. Concentratable Entanglement (CE) for an $N$-qubit pure state $|\psi\rangle$ is then defined by averaging over the purities of all possible subsystems of a given size $k$:

$$
\mathrm{CE}_k(|\psi\rangle) = \frac{2^k}{2^k - 1}\left(1 - \frac{1}{\binom{N}{k}}\sum_{|S|=k}\mathrm{Tr}(\rho_S^2)\right) = \frac{2^k}{2^k - 1}\cdot\frac{1}{\binom{N}{k}}\sum_{|S|=k}\left(1 - \mathrm{Tr}(\rho_S^2)\right).
$$

In this equation, the sum is taken over all $\binom{N}{k}$ possible subsystems $S$ of size $k$. $\rho_S$ represents the reduced density matrix of the subsystem $S$. $1-\mathrm{Tr}(\rho_S^2)$ measures how mixed the subsystem is, with a larger value implying greater entanglement. The entire expression is then averaged and normalized. The purity term $\mathrm{Tr}(\rho_S^2)$ for each subsystem can be estimated on a quantum computer using the **SWAP test** (further explained below). The SWAP test requires two copies of $\rho_S$ and measures the expectation value of the SWAP, which directly corresponds to the purity of the state. Consequently, estimating CE involves preparing two copies of the global state and performing SWAP tests on all corresponding subsystems of size $k$.

Estimating the CE value for a given state thus requires testing the purity of every possible subset of qubits in the state, which makes computing this value intractable as the qubit count increases. This motivates the use of efficient quantities that preserve the ordering and distributional structure of CE. We thus use two measurement-efficient quantities that are connected to CE and straightforward to obtain on current hardware.

For training circuits to estimate CE distributions (as in Fig. 3), we use

$$
\mathrm{NZP} \;=\; 1 - P(0^n),
$$

i.e., the complement of the all-zeros outcome in the computational basis. NZP is a lightweight coherence indicator that increases as probability mass spreads away from a basis state. We use NZP as a cheap surrogate during optimization where more precise CE estimators would be too expensive. Estimating NZP is scalable because, with shallow ansatzes, the probability of observing the all-zero outcome does not vanish exponentially, reducing NZP estimation to a Bernoulli mean estimation problem with shot complexity $O(\epsilon^{-2})$ independent of qubit count. NZP is used solely as a CE surrogate rather than for state reconstruction, allowing the overall cost to remain polynomial and compatible with near-term hardware.

Our circuits that estimate the soil moisture sensors use a more accurate and expensive measure to estimate the CE value by leveraging a subset of SWAP tests (namely, only on single-qubit pairs) that are used to generate CE estimations. We prepare two copies of the state and run parallel SWAP tests on single-qubit subsets $S = \{j\}$. Let $q = \Pr[\text{all SWAP ancillas} = 0]$ and $n$ be the number of data qubits. Then,

$$\frac{4}{n}\big(1 - q\big) \;\leq\; \mathrm{CE}_1 \;\leq\; 4\big(1 - q\big),$$

where $\mathrm{CE}_1 = 4\big(1 - \frac{1}{n}\sum_{j=1}^{n} p_{0,j}\big)$ and $p_{0,j} = \Pr[\text{ancilla } j = 0] = \frac{1+\mathrm{Tr}(\rho_j^2)}{2}$. This bound is conservative and equals $\mathrm{CE}_1$ when single-qubit purities are equal or else safely overestimates $\mathrm{CE}_1$.

**Scalability.** Estimating CE precisely does not scale since it requires aggregating purities over all size-$k$ subsystems of qubits in a single state, and thus needs SWAP-test–based purity estimation on a combinatorial number of subsets, which becomes intractable as the number of qubits grows. Looking ahead to error-corrected quantum computing, scalability becomes even more critical since a single logical qubit typically uses $O(d^2)$ physical qubits and continuous syndrome cycles, and thus any metric whose evaluation cost grows superlinearly in the number of logical qubits is completely unscalable. The measurement-efficient quantities we use above are designed to circumvent this issue.

During training, we use our lightweight surrogates without paying the full evaluation cost; for soil moisture evaluation, we use the single-qubit, two-copy procedure that prepares two copies and runs SWAP tests in parallel on $S = \{j\}$, aggregates those local outcomes, and then relates the aggregate to CE via the established bounds above. Using this, our evaluation cost grows with the number of local SWAP tests we choose to run, proportional to $n$ when we test each qubit once in parallel, instead of with the number of subsets of qubits. That keeps shot budgets linear, which is compatible with near-term hardware. Empirically, our evaluations demonstrate that this pipeline maintains stable ordering and trends under noise, confirming that these metrics remain informative when direct CE estimation is infeasible.

## A.2 SWAP Test Metric

Given two $n$-qubit registers and an ancilla initialized to $|0\rangle$, the SWAP test applies a Hadamard, a controlled-swap on some subset $S \subseteq \{1, \ldots, n\}$ of corresponding qubits, and a final Hadamard to the ancilla. Measuring the ancilla yields:

$$p_0(S) \;=\; \Pr[\text{ancilla} = |0\rangle] \;=\; \tfrac{1}{2}\left(1 + \mathrm{Tr}[\rho_S \, \sigma_S]\right),$$

where we get $|0\rangle$ more often when the states overlap more. Thus in practice, $p_0(S)$ can be used as a similarity score where:

$$p_0(S) \approx 1 \;\Rightarrow\; \text{the two states are nearly the same,} \qquad p_0(S) \approx \tfrac{1}{2} \;\Rightarrow\; \text{they are nearly orthogonal.}$$

Moreover, the SWAP test can also be used to compute the purity of a given state; in fact, the ancilla's measurement encodes the purity of the subsystem $S$ of a single copy when the two inputs are identical ($\rho = \sigma$). Intuitively, the more often we see $|0\rangle$, the more pure $S$ is on its own, meaning it carries little correlation with the rest of the system, while outcomes closer to $1/2$ indicate $S$ is mixed because its information is shared with (i.e., entangled with or randomized by) its complement.

## B QUANTUM SENSOR SIMULATOR CIRCUITS

This appendix provides a description of the quantum circuits used to simulate the soil moisture and dark matter quantum sensing protocols. These circuits are adapted from the STQS framework (Jebraeilli et al., 2025) and are designed to model the specific physical interactions relevant to each application.

### B.1 SOIL MOISTURE SENSOR CIRCUIT

The circuit for the soil moisture sensor is designed to perform a differential measurement, comparing a signal reflected from the soil to a reference signal from free space. The purpose of this protocol is to determine the soil's dielectric permittivity, which is directly correlated with its moisture content. The structure of the circuit begins by preparing a set of sensor qubits into a Greenberger-Horne-Zeilinger (GHZ) state, entangling the qubits. Following state preparation, the entangled qubits are partitioned into two groups. The first group interacts with the target signal, accumulating a phase $\phi_{\text{soil}}$, while the second group interacts with the reference signal, accumulating a phase $\phi_{\text{free}}$. The resulting phase difference, which contains the information about the soil moisture, is then transferred onto a single memory qubit using a sequence of CNOT gates. Finally, the sensor qubits are measured in a disentangled basis, using entanglement to amplify the small phase difference between the two signals. A circuit diagram for the soil sensor can be found in Fig. 8 in Jebraeilli et al. (2025).

### B.2 DARK MATTER SENSOR CIRCUIT

The circuit simulating the dark matter detector is designed to sense a faint, oscillating signal hypothesized to originate from ultralight, wavelike dark matter. The goal is to achieve a high degree of sensitivity to detect a weak interaction. The protocol starts by preparing an array of sensor qubits in an entangled GHZ state, which acts as a collective probe. The sensing phase is modeled by applying a small rotation, represented by an $R_x(\phi)$ gate, to each of the sensor qubits simultaneously. The rotation angle $\phi$ is proportional to the strength of the interaction with the dark matter field. The use of an entangled array provides a coherent amplification of this weak signal, as the effect of the rotation on the collective state is more pronounced than on single unentangled qubits. After the interaction, disentangling gates are applied to transfer the accumulated phase information from the sensor array to a single qubit. This information is then mapped to a memory qubit for measurement. A circuit diagram for the dark matter sensor can be found in Fig. 15 in Jebraeilli et al. (2025).

## C LLM USAGE

ChatGPT and Google Gemini were used to help generate/refine code, as well as refine paper content. All generated content was checked by the authors for correctness.

## D REPRODUCIBILITY STATEMENT

QMILL's data generation methodology, machine learning codebase, and generated datasets are open-sourced at: *https://anonymous.4open.science/r/QMill-FA93*. This ensures transparency and reproducibility, supporting research acceleration.

