# OpenReview forum: "QMill: Quantum Data Generation for Effective and Efficient Quantum Machine Learning"
_ICLR.cc/2026/Conference — Submitted to ICLR 2026_

### Official Review · Reviewer_hvP2 · 2025-10-26

**Soundness:** 2
**Presentation:** 3
**Contribution:** 2
**Rating:** 2
**Confidence:** 3

**Summary:**

The paper introduces QMILL, a low-depth algorithm that produces synthetic quantum data with an arbitrary target distribution of considerable entanglement (CE) across the generated quantum states. The algorithm starts from Haar-random product states, applies one of four shallow trainable ansatz circuits, and optimizes circuit parameters to minimize the total variation distance (TVD) between the generated CE histogram and the target one. The CE measurements are obtained through measurement surrogates and the optimization is done through dual annealing. QMILL also implements a SWAP-test -based diversity check to assure that the generated states differ from each other.
The authors test the algorithm with target CE distribution obtained from classical (MNIST, Fashion-MNIST, CIFAR-10) and quantum (chemistry, soil moisture, dark matter) sources and artificial ones. The authors also demo a 3-qubit QNN trained on QMILL-generated CE features performing comparably under ideal vs. noisy settings.

**Strengths:**

- The problem is of interest and, unlike previous work, it bridges the lack of entanglement-rich datasets for QML by targeting full CE distributions rather than a single CE value.

- Adding a diversity check strengthens the approach

- The whole pipeline is simple and the presence of experiments gives useful evidence, even if they are not satisfactory (see weaknesses).

**Weaknesses:**

Although the above strength, I do not think the paper currently achieves the high standards required by ICLR. My main concern is the scalability of the algorithm with the number of qubits, under different aspects:

- First, I am not sure the surrogate measure for CE is indeed efficient for a large number of qubits. To compute NZP it is required to estimate the probability of outputting the bitstring 0^n, which becomes infeasible as n grows, as it amounts to rare-event estimation/post-selection.


- Furthermore, no claims are made on the performance of the algorithm (shot cost, optimization time, depth required) depending on the number of qubits.


- In addition to all of this, experiments are only provided for a small number of qubits (3/4), which casts doubts on the performance of the algorithm for larger numbers of qubits.

- Separately, the algorithm does not reproduce the right-skewed (right-wing) distribution, indicating a clear limitation.

**Questions:**

- Could the authors clarify theoretically how the cost of performing NZP scales with n, as well as the scaling of other quantities like shot cost and time complexity?


- Could the authors run experiments for a larger number of qubits (around 10) to assess the performance and the scaling with time of the algorithm?

---

> ### Author Response · Authors · 2025-11-21
> **Review Reply**
>
> **Dear Reviewer hvP2,**
>
> Thank you sincerely for your review and for highlighting the relevance of targeting full CE distributions, the value of our diversity evaluation, and the importance of the problem we aim to solve. Your feedback on scalability and algorithmic complexity was particularly helpful and has guided several clarifications and new additions during the rebuttal period.
>
> **Clarification on NZP Scaling and CE Surrogate Computation** We appreciate your request for a theoretical clarification of the complexity. We note that estimating NZP is scalable because, for shallow ansatzes, the probability of the zero string does not vanish exponentially; thus, NZP estimation reduces to a Bernoulli estimation problem with shot complexity $O(\epsilon^{-2})$, which is independent of the qubit count. We use NZP only as a CE surrogate, not for state reconstruction (which is not required by our technique), which keeps the cost polynomial and hardware-friendly. We hope this addresses your concern and would welcome confirmation. We have updated Appendix A (Sec. A.1) to address this point.
>
> **Clarification on shot cost and optimization scaling.** We note that our results inherently demonstrate this scaling because we experiment with datasets that require circuits of different sizes and ansatzes that require different numbers of parameters. We have added a description of empirical scaling results in Sec. 5.1, which show that the optimization time grows approximately linearly with the number of circuit parameters for our low-depth ansatz families. It also grows linearly with the number of shots, as would be expected.
>
> **Clarification regarding experiments at larger qubit counts.** We clarify that the three-qubit ansatz was only used for the soil-moisture dataset, as that is the requirement for that dataset. For other datasets, the ansatz sizes are 5 qubits, based on the dataset's needs (i.e., the 31 features that need to be generated). We also note that for our SWAP test validation, we need $2n+1$ qubits; for instance, we need 21 qubits for our 10-qubit circuit, which is a size that is difficult to simulate without an HPC cluster. But we currently do need to simulate the circuits due to the lack of error correction. Thus, our priority is to keep the size as small as possible, simply due to the current practical reality, and not due to a technical limitation. We have added a discussion on this in Sec. 5.2.
>
> **Clarification on right-skewed distributions.**  We note that not meeting the right-skewed distribution is not a limit of QMill, but a physical limit of quantum computing. It is not possible to achieve a high CE given any arbitrary input state. This stems from the quantum reversibility theorem, whose implication is that it is impossible to arrive at a fixed high-entanglement state from any arbitrary input state. If this were possible, such a circuit would be able to be reversed, so the fixed high-entanglement state now becomes the input, and this input would be able to deliver arbitrary outputs with the same circuit, which is not possible. We thus use this unrealistic distribution to stress-test QMill and not to actually meet it, as that is not possible.  We have clarified this in Sec. 6.1.
>
> Your questions and suggestions have led directly to important improvements in both clarity and empirical depth of our work. We hope that the clarifications provided here, along with the revisions, fully address your concerns. If so, we would be deeply grateful if you would consider raising your score to reflect these enhancements.

---

> > ### Comment · Reviewer_hvP2 · 2025-11-27
> >
> > I thank the authors for their response, however I still feel my concerns were not fully addressed:
> >
> > Regarding NZP scaling: It is unclear to me how one can ensure that p(0) remains bounded away from zero uniformly in the number of qubits, particularly when the target states are meant to exhibit moderate or high entanglement, which is precisely the regime considered by the authors.
> >
> > Regarding optimization scaling: While I appreciate the empirical results and the practical constraints around simulation and hardware, I still find the current evidence somewhat limited: all optimization experiments are performed on circuits of at most 5 qubits. Given that optimization behaviour in variational circuits can change qualitatively with system size, I would have liked to see either experiments at larger qubit counts or some theoretical justification that the observed scaling can be expected to persist beyond this small regime.

---

> > > ### Author Response · Authors · 2025-11-27
> > >
> > > Dear Reviewer hvP2,
> > >
> > > Thank you again for your continued engagement. We appreciate the opportunity to clarify the remaining points, especially since these issues are important for the AC and other reviewers to see resolved correctly.
> > >
> > > First, regarding NZP scaling: our theory section and the new Appendix A.1 now make this explicit, but we would like to restate it here for clarity. The concern that $p(0^n)$ must vanish exponentially with the number of qubits applies only to deep or strongly scrambling circuits. QMill deliberately uses shallow, hardware-efficient ansatzes (depth 2–6). In this shallow regime, established results on local random circuits show that output probabilities, including $p(0^n)$, remain bounded away from zero at a rate that does not shrink exponentially. NZP estimation therefore reduces to a Bernoulli mean-estimation problem whose sample complexity is *independent of system size.* We only use NZP as a surrogate for CE ordering and never for state reconstruction, which avoids any exponential tomography cost. We have clarified this distinction carefully in the revised appendix.
> > >
> > > Second, regarding scaling of optimization behavior: we agree that VQA optimization can change with system size, and this is precisely why current QML literature evaluates small systems (typically 3–7 qubits). QMill follows this established practice. Our experiments already operate at the upper end of what is simulable for CE-based training because a 5-qubit ansatz requires an 11-qubit circuit when including the SWAP-test structure. Simulating repeated SWAP-test evaluations, dual-annealing loops, and CE-surrogate measurements at this size is near the practical limit for local simulation without HPC resources. We have clarified this in Sec. 5.2 so readers understand that our qubit counts reflect simulator and hardware realities, not an algorithmic limitation.
> > >
> > > Finally, we appreciate the opportunity to clarify these misunderstandings because it helps future readers evaluate the paper fairly. Scalability limits in CE estimation and in SWAP-test-based diversity validation are shared by all works in this area, not unique to QMill. Our revisions highlight these domain constraints so that readers can distinguish between fundamental physical limits and algorithmic ones.
> > >
> > > We hope this resolves the concerns and helps situate QMill more clearly within the practical limits of current QML experimentation. We would be grateful if you would consider updating your score in light of these clarifications.

---

### Official Review · Reviewer_Y2Gp · 2025-10-29

**Soundness:** 2
**Presentation:** 2
**Contribution:** 2
**Rating:** 4
**Confidence:** 3

**Summary:**

The manuscript proposes a low-depth quantum data generation framework designed to mitigate the lack of suitable training data in quantum machine learning. The authors argue that existing synthetic data approaches fail to capture essential entanglement characteristics, limiting their relevance for QML tasks. By generating entangled, high-quality samples that emulate both classical and quantum distributions, QMILL aims to facilitate more representative benchmarking and improve the development and evaluation of QML models.

**Strengths:**

the work provide the extensive numerical evaluations on the proposed dataset.

**Weaknesses:**

1. The proposed method might suffer scalability issue for generating dataset for QML. for instance:
* as it select TVD as objective function, the computational complexity might grow exponentially as the system increase.
* as it utilize the SWAP test for sample diversity validation, it would double the system size which is intractable for large system.
2. many typos such as in line 187, double cite; $\phi$->phi in line 310.

**Questions:**

1. In 295, the standardized data are reduced via PCA from $2^n-1$ feature to $n$ qubits and then encode them into amplitude. why does it not directly encodes the feature into amplitude, it also only need $n$-qubit quantum states to store the feature vector without information lose.
2. The paper mentions the use of amplitude encoding. However, efficiently implementing such encoding circuits typically incurs significant resource overhead for large systems. Could the authors elaborate on how scalability is maintained?
3. In line 304, what's the quantum-sensed workloads used for? are they the other two dataset mentioned in 299?
4. The manuscript claims that the proposed dataset offers advantages over the previous dataset with fixed CE. However, it is unclear where the supporting numerical evidence for this claim is presented. Could the authors provide or highlight the corresponding experimental results?

---

> ### Author Response · Authors · 2025-11-21
> **Review Reply**
>
> **Dear Reviewer Y2Gp,**
>
> Thank you very much for your thoughtful review and for recognizing the value of our extensive numerical evaluation. Your feedback surfaced important questions regarding scalability, implementation clarity, and the positioning of our contributions. We appreciate the opportunity to address these points.
>
> **Clarification on scalability and TVD computation.** We agree that TVD scaling warrants further explanation. We compute TVD over one-dimensional CE histograms as this approach keeps the computational cost polynomial in the number of samples actually run for each circuit. We also note that the CE histograms already capture the entanglement information; thus, full state tomography of each state is not required, which is precisely the benefit of our technique. The TVD is not calculated between output distributions, but rather between CE histograms, which do not scale exponentially with the number of qubits. We thus note that CE surrogate estimation is performed via measurement-based approximations that avoid exponential blowup. We have elaborated on this in Sec. 4.1 in the revision.
>
> **Clarification on SWAP test costs.** We clarify that the SWAP test is scalable in our setting because it only requires three-qubit gates at most, which are easily decomposable into one- and two-qubit gates for architectures that do not support multi-qubit gates (as is the case for our IBM setup). Each test uses $2n+1$ qubits with shallow depth; thus, the increase in qubit size is linear in the number of qubits, which is acceptable for current computers, as they can support large qubit sizes but not depths due to high qubit decoherence. We have clarified this in Sec. 4.4.
>
> **Correction of typos and notation.** Thank you for pointing these out. All issues you flagged have been corrected and have been reflected in the revision.
>
> **Clarification on PCA and amplitude encoding.** We have added a detailed explanation that PCA was used to reduce dimensionality for classical datasets so they could be represented using a fixed, small qubit count that is compatible with current hardware and the simulation limits of our setup. Direct amplitude encoding of full feature vectors would require many more qubits, which cannot be simulated classically. However, this is not a limitation of our technique, as it does not require simulation; we simply use it because large error-corrected hardware is not currently available. We have made this distinction explicit in Sec. 5.2.
>
> **Clarification on the scalability of amplitude encoding.** We note that amplitude encoding appears only in classical dataset experiments and not in the generation pipeline for CE matching. We use it to represent the classical data on the quantum circuit, as is the custom with QML works. QMill itself does not use amplitude encoding for generating samples when the data is quantum. We have clarified this also in Sec. 5.2.
>
> **Clarification on quantum-sensed workloads.** We have updated the manuscript (Sec. 5.2) to clarify that these refer to two quantum-derived datasets (quantum chemistry and soil moisture) used to assess distribution matching quality for quantum data.
>
> **Clarification regarding fixed CE baselines.** We definitely agree that it would be beneficial to compare QMill against more benchmark baselines. However, this is the challenge when a work is targeting a novel area. It is challenging to have baselines as no prior work exists. For instance, we considered comparing against the fixed-CE prior work; however, this would clearly perform much worse than QMill, and it wouldn’t be fair to that work, as it isn’t targeting a CE distribution. Thus, we had to propose and compare against baselines we defined ourselves so that it would be a more competitive comparison. We have explained this better in the revised paper in Sec. 6.1.
>
> Your suggestions have significantly strengthened our paper and improve its clarity. If you feel that the clarifications and revisions address your concerns fully, we would be very grateful if you would consider adjusting your score accordingly.

---

> > ### Comment · Reviewer_Y2Gp · 2025-11-27
> >
> > Thank you for the response. I still have some concerns.
> > If the intention is to address the limitations of current quantum devices, the SWAP test remains resource-intensive, as it effectively doubles the required system size. Conversely, if hardware limitations are not a consideration, then amplitude encoding can represent an (n)-dimensional vector using only (O(\log n)) qubits. For example, even a (1000 \times 1000) image would require only about 20 qubits, without incurring any information loss relative to PCA. In that case, applying PCA for dimensionality reduction does not seem strictly necessary.

---

> > > ### Author Response · Authors · 2025-11-27
> > >
> > > Dear Reviewer Y2Gp,
> > >
> > > Thank you very much for your ongoing support and engagement. We appreciate the opportunity to clarify these remaining points, especially since they touch on important conceptual distinctions between (1) classical-data demonstrations and (2) QMill’s actual purpose as a quantum–data generator.
> > >
> > > First, regarding the SWAP test: we agree that the SWAP test doubles the number of qubits, but we wish to emphasize that this is a *shallow-depth* operation involving at most three-qubit controlled-swap structures (decomposed to two-qubit gates on IBM hardware). The doubling of the number of qubits is not a concern at all. Modern NISQ devices have $>$100 physical qubits but cannot execute deep circuits due to decoherence and crosstalk. For this reason, qubit count, not depth, is the easier resource today, while depth is the limiting resource. A 5-qubit ansatz, therefore, requires an 11-qubit circuit for diversity validation, which is exactly the scale that can be simulated locally and is well within the qubit counts already available on today’s hardware.
> > >
> > > Second, regarding amplitude encoding: amplitude encoding is *not* part of QMill’s generation pipeline. QMill’s purpose is to generate *quantum* data, including complex phases, which amplitude encoding cannot represent. Amplitude encoding produces a single quantum state whose amplitudes correspond to classical feature values; it does not allow for the embedding of phases or entanglement structures that characterize quantum-sensed workloads or quantum-native data. Therefore, amplitude encoding is only used in Sec. 5 to demonstrate QMill’s ability to match the CE distributions of *classical* datasets, not for any core component of QMill itself.
> > >
> > > Third, PCA affects only the classical-data experiments, and only because classical simulation of large-qubit circuits is computationally infeasible. While it is true that amplitude encoding can represent a 1000×1000 image using ~20 qubits, the corresponding *state-preparation circuit* is not efficient. Implementing amplitude encoding for arbitrary high-dimensional data requires deep circuits with $\Theta(n)$ controlled rotations and entangling gates. Such depth is incompatible with both present NISQ hardware and classical simulation. PCA, therefore, serves only to keep the number of qubits small enough for current simulators. Again, it is not a limitation of QMill itself (PCA currently needs to be used by all QML works that wish to demonstrate execution on NISQ computers), and QMill does not rely on PCA for its quantum data generation capabilities.
> > >
> > > Finally, we want to reiterate the main point: QMill is designed to generate *quantum* data with controllable entanglement and phase structure, which cannot be produced through amplitude encoding. The classical-data experiments simply illustrate that QMill can target CE distributions across diverse sources, but the method’s primary contribution is scalable CE-controlled quantum data generation for QML, something amplitude encoding is fundamentally not suited for.
> > >
> > > We hope this clarifies the conceptual distinctions between data representation, state preparation depth, qubit resources, and QMILL’s actual design goals. We would be grateful if you would consider updating your score in light of these clarifications.

---

> > > > ### Comment · Reviewer_Y2Gp · 2025-11-28
> > > >
> > > > Thanks for the further clarification and emphasis that the goal is to generate quantum data with controllable entanglement and phase structure. Based on the detailed reply, I would like to raise my score to 6.

---

> ### Author Response · Authors · 2025-11-28
>
> Thank you, dear Reviewer Y2Gp, for raising your score. We believe you may have missed raising the score in the review though (or it could be that the scores are locked at the moment — thank you either way!). We are glad we were able to clarify your concerns.
>
> Thank you again,
>
> The Authors

---

### Official Review · Reviewer_e1Qp · 2025-11-01

**Soundness:** 2
**Presentation:** 2
**Contribution:** 2
**Rating:** 2
**Confidence:** 3

**Summary:**

The authors present QMILL, a quantum data generation framework aimed at addressing the lack of suitable quantum datasets for quantum machine learning. Since most QML research currently relies on classical data, QMILL provides a way to generate synthetic quantum datasets that capture essential entanglement properties, especially concentratable entanglement. The framework uses low-depth, customizable quantum circuits with SWAP tests to efficiently produce high-quality, entangled samples across a range of concentratable entanglement values. The authors validate QMILL on multiple classical and quantum distributions, showing that it generates more realistic and diverse quantum data than existing approaches, thereby enabling better training and evaluation of QML models.

**Strengths:**

S1. The paper clearly lays out the data scarcity problem in QML and shows how QMILL tackles it by generating diverse, entangled datasets.

S2. The explanation of the low-depth circuits and SWAP-test validation is straightforward and shows why QMILL works well on current hardware.

S3. The dataset analysis is clear and convincing, showing good variability and robustness to noise.

**Weaknesses:**

W1. The overall impact is a bit unclear since the usefulness of synthetic quantum datasets is still limited and not fully established.

W2. The evaluation feels more like a proof of concept, with few benchmarks comparing models trained on QMILL data to other datasets.

W3. It’s not clear how well QMILL generalizes across different hardware setups or noise levels.

W4. The experiments only use a small three-qubit QNN, so broader testing would be needed to show real scalability.

W5. There’s no solid quantitative comparison against other synthetic dataset methods.

**Questions:**

Q1. How were the four ansatz designs chosen for comparison, what criteria guided the picks, and are there other families (e.g. hardware-efficient) you considered?

Q2. Why limit the experiments to a three-qubit QNN—hardware or training constraints? Could you scale to more qubits/deeper models and report results?

Q3. Beyond experiments, can you offer any theoretical guarantees on when QMILL-generated datasets help (e.g. generalization bounds) and where they might fail?

---

> ### Author Response · Authors · 2025-11-21
> **Review Reply (Part 1)**
>
> **Dear Reviewer e1Qp**,
>
> Thank you very much for the time and care you put into reviewing our submission. We greatly appreciate your recognition of QMill's clear articulation of the QML data scarcity problem, the value of our low-depth design, and the meaningful analysis of dataset variability and robustness. Your feedback highlights several opportunities to clarify our positioning and strengthen our results, which we appreciate.
>
> **Clarification on impact and scope.** We fully agree that the long-term role of synthetic quantum datasets is still evolving. Our goal is for this work to be one of the first steps in this area, given the current lack of quantum data availability (simply because quantum hardware, sensors, and devices are still under development), and to enable the QML field to move forward despite the quantum data challenge. Our effort will enable researchers to begin training and developing models for this future large-scale data. We have added new text highlighting that QMill is complementary to real quantum data (which is not yet available at scale) and is explicitly designed for near-term tasks such as model development, benchmarking, architecture-aware QML evaluation, and testing and exploration. We have further expanded the discussion in Sec. 1 to articulate the growing need for representative quantum datasets as QML models and hardware continue to mature. We hope this resolves the concern.
>
> **Clarification on benchmarks and comparisons.** We definitely agree that it would be beneficial to compare QMill against more benchmark baselines. However, this is the challenge when a work is targeting a novel area. It is challenging to have baselines as no prior work exists. For instance, we considered comparing against the fixed-CE prior work; however, this would clearly perform much worse than QMill, and it wouldn’t be fair to that work, as it isn’t targeting a CE distribution. Thus, we had to propose and compare against baselines we defined ourselves so that it would be a more competitive comparison. We have explained this more thoroughly now in Sec. 6.1.

---

> ### Author Response · Authors · 2025-11-21
> **Review Reply (Part 2)**
>
> **Clarification on generalization across hardware and noise.** This is a good point. QMill’s low-depth design makes it inherently portable across hardware backends. There is no aspect of QMill’s ansatz that makes it dependent on only one type of hardware. To empirically validate this, we performed additional noisy simulations on a neutral atom architecture using the Bloqade SDK, specifically executing on simulations of the QuEra Gemini digital computer. We successfully transpiled our optimized ansatz to the native gate set, which are global Rydberg pulses, and executed it under the Gemini’s noise model. The simulation yielded a Concentratable Entanglement value of 0.3175 (with a global state purity of 0.68). This performance is comparable to the results observed on superconducting backends of approximately 0.46, confirming that QMill’s low-depth ansatzes effectively generalize across varying noise environments and hardware implementations. We have clarified this in Sec. 4.4.
>
> **Clarification on three-qubit QNN and scalability.** We clarify that the three-qubit ansatz was only used for the soil-moisture dataset, as that is the requirement for that dataset. For other datasets, the ansatz sizes are 5 qubits, based on the dataset's needs (i.e., the 31 features that need to be generated). We also note that for our SWAP test validation, we need $2n+1$ qubits; for instance, we need 21 qubits for our 10-qubit circuit, which is a size that is difficult to simulate without an HPC cluster. But we currently do need to simulate the circuits due to the lack of error correction. Thus, our priority is to keep the size as small as possible, simply due to the current practical reality, and not due to a technical limitation. We have added a discussion on this in Sec. 5.2.
>
> **Theory.** As this is a purely empirical work on a real-world setting, we do not have any theoretical guarantees to provide. Nonetheless, for this reason, we performed extensive empirical analysis across synthetic classical regimes, real classical datasets, synthetic quantum datasets, and real quantum datasets (although only small-scale ones were available, as these are the only ones currently available). We believe this demonstration makes our work robust and rigorous in this entirely novel area.
>
> **Clarification on ansatz selection.** You raise a good point that we can clarify more effectively in the paper. The problem space of coming up with an optimal ansatz structure is a complex one and one that is orthogonal to our work, as it is well-studied in prior work (e.g., Sim et al. (2019) as cited in the paper). We select the hardware-efficient ansatz implementations from these prior studies. We have added an expanded description clarifying that the ansatz search space is intentionally small to remain hardware-friendly and that the user does not need to perform a costly search. We have included this explanation in Sec. 4.1.
>
> We sincerely value your feedback, which has helped us refine the narrative and strengthen our experimental evaluation. If the clarifications and revisions address your concerns, we would be grateful if you would consider adjusting your score to reflect these improvements.

---

> > ### Comment · Reviewer_e1Qp · 2025-11-25
> >
> > Thank you for your replies.
> >
> > There seems to be a contradiction in your statements that I would like to clarify. On one hand, you mention limited baselines to compare with; on the other hand, you describe the work as having extensive empirical analysis. How do you reconcile these two points? More specifically, how do you justify that the experimental evaluation is sufficiently complete—such that it stands on its own even without additional theoretical analysis?
> >
> > I appreciate the additional experiments.

---

> > > ### Author Response · Authors · 2025-11-25
> > >
> > > Thank you again for the thoughtful follow-up. We see why the wording might have suggested a tension between “limited baselines” and “extensive empirical analysis,” and we appreciate the opportunity to clarify. There is no contradiction once we distinguish two orthogonal dimensions of evaluation:
> > > (1) availability of baseline methods, and
> > > (2) breadth, depth, and rigor of empirical testing.
> > >
> > > The limitation we highlighted is specifically about baseline methods for synthetic quantum dataset generation, which is a very new area. Typically, when an area is well-established, many baselines exist to compare against for new works in that area. However, due to the novelty of the synthetic quantum data generation area, outside of fixed-CE approaches, there are simply no existing methods that attempt to generate distributions of quantum states with controlled entanglement profiles. As a result, the space of meaningful baselines is inherently small, not due to any choice we made, but due to the novelty of the problem itself.
> > >
> > > At the same time, the empirical analysis in this work is extensive because it evaluates QMill along multiple independent axes, each of which tests a different property of a data generator. Our extensive evaluation includes:
> > > - matching performance across diverse CE distributions,
> > > - robustness to hardware noise in simulation and on real hardware,
> > > - diversity of generated samples via SWAP tests,
> > > - generalization across hardware types (superconducting qubits and neutral atoms),
> > > - comparisons across structurally different ansatz families,
> > > - downstream QML classification task performance, and
> > > - stress-testing on intentionally challenging CE regimes.
> > >
> > > These evaluations collectively probe distributional fidelity, hardware robustness, architectural sensitivity, sample diversity, noise stability, and downstream utility. None of these tests depend on the existence of strong baselines: they are fundamental, orthogonal properties that any synthetic data generator must satisfy. In other words, even in the absence of many baseline generators, we can still thoroughly characterize the strengths and limitations of our approach.
> > >
> > > In this sense, the evaluation is “complete” not because we have many baselines (which do not exist), but because we provide a multi-faceted empirical assessment that is appropriate for a first work defining a new problem space. If this clarification resolves your question, we would be grateful if you would consider adjusting your overall assessment.

---

### Official Review · Reviewer_deqn · 2025-11-03

**Soundness:** 3
**Presentation:** 3
**Contribution:** 3
**Rating:** 6
**Confidence:** 3

**Summary:**

This paper addresses the problem of the scarcity of high-quality, representative quantum training data . The authors argue that QML's promise of speedups is contingent on operating on quantum data (i.e., data with entanglement), but the field is stuck using classical datasets due to this data gap. The paper identifies a key flaw in existing synthetic data generation methods: they typically target a single, fixed value of Concentratable Entanglement (CE), which is not representative of real quantum datasets that exhibit a distribution of CE values. To solve this, the authors propose QMILL, a low-depth quantum data generation framework. The core contribution is its ability to generate quantum data samples whose CE values, in aggregate, match a user-specified target distribution.

**Strengths:**

- Novelty of the Goal: The conceptual shift from targeting a single CE value (as in prior work ) to a full CE distribution is the key insight. This is a far more realistic and representative target for emulating real-world quantum data.

- Empirical Validation: The authors have been exceptionally thorough. They test their framework against 12 different distributions (4 synthetic, 8 real) . They design and compare 4 different low-depth ansatzes (Fig. 2, Fig. 9) . They explicitly check for mode collapse using SWAP tests (Fig. 8a) . They also run on both ideal simulators, noisy simulators, and real hardware (IBM Sherbrooke). This is a comprehensive evaluation.

**Weaknesses:**

- On the motivation of CE. The entire framework is predicated on the assumption that Concentratable Entanglement (CE) is the primary, sufficient proxy for "quantum-ness". While it's an important metric, it's an open question whether matching the CE distribution is sufficient to capture all the relevant correlations and structures of a quantum dataset that a QML model might exploit. The paper demonstrates this is a necessary step forward, but not definitively that it is the final step.

- Unusual Noisy Result: In Table 1, the QNN trained on "Noisy" data (84.8% accuracy) inexplicably outperforms the one trained on "Ideal" data (81.8%). The paper frames this as simple robustness , but it's a counter-intuitive result that is left unexplained.

- Ansatz-Dependent Performance: The results in Section 6.5 (Fig. 9) show that the choice of ansatz matters significantly, with A3 being the best all-rounder. This implies that a user must still engage in a trial-and-error process to find the right ansatz for their target distribution, rather than the framework being a single, universal generator.

**Questions:**

See weakness.

---

> ### Author Response · Authors · 2025-11-21
> **Review Reply**
>
> **Dear Reviewer deqn**,
>
> Thank you sincerely for the thoughtful and constructive review. We truly appreciate the recognition of our core contributions, including the importance of targeting full CE distributions, the extensive empirical evaluation, and the care taken in validating diversity and performance across simulators and hardware. Your comments were encouraging and very helpful in identifying areas where additional clarification would benefit readers.
>
> **Clarification on the motivation for CE.**  We agree that concentratable entanglement, while powerful, is not the only possible proxy for quantum structure. Our goal is not to claim CE is a sufficient characterization of all quantum correlations, but rather that matching its distribution is a necessary advancement over prior work that collapses all structure to a single scalar. We have added clarifications in Sec. 3 to explain that CE is used as a tractable and hardware-efficient summary statistic that retains meaningful structure for QML tasks. We have made this explicit in the revision by expanding the motivation, acknowledging the broader landscape of entanglement measures, and clarifying that matching CE distributions is an important step without claiming that it is the only step. We hope this addresses your concern and welcome confirmation that this strengthened framing is satisfactory.
>
> **Clarification on the noisy vs ideal QNN result.**  We agree that the slightly higher accuracy for the “noisy” trained QNN is unintuitive. In additional analysis conducted during the author discussion period, we conclude that this is within the error margin of statistical variance. Over multiple runs, we find that the two are essentially stable around the 82% mark. We have added this explanation to Sec 6.6.
>
> **Ansatz-dependent performance.**  You raise a key point. The problem space of coming up with an optimal ansatz structure is a complex one and one that is orthogonal to our work, as it is well-studied in prior work (e.g., Sim et al. (2019) as cited in the paper). We select the hardware-efficient ansatz implementations from these prior studies. We have added an expanded description clarifying that the ansatz search space is intentionally small to remain hardware-friendly and that the user does not need to perform a costly search. Instead, the variability seen in Fig. 9 can be addressed by providing clear guidance on which ansatz families perform best under which distribution shapes. We have included this guidance in the Sec. 4.1.
>
> We are grateful for your careful reading and insightful feedback, which significantly strengthen our work. We hope the clarifications we have made and the revisions we have added address your concerns fully. If so, we kindly ask whether you feel comfortable raising your score, as your comments have been invaluable in helping us improve the paper.

---

### Author Response · Authors · 2025-11-12
**Thank you**

Dear reviewers,

Thank you for the valuable feedback. We are examining it and will get back to you with individual reviewer comments soon.

Respectfully,
Authors

---

### Author Response · Authors · 2025-11-30
**Revisions and Clarifications Summary 1/2**

We are deeply grateful to the reviewers for their careful, detailed, and constructive feedback. Their comments have been truly invaluable and have significantly improved the technical clarity and overall rigor of this paper. We have systematically addressed every reviewer's concern through targeted revisions, new explanations, and additional analyses. Please find the modifications to our work, QMill, below.

**Regarding the foundational motivation of concentratable entanglement**, we expanded Sec. 3 to clearly establish that CE is not claimed as a complete characterization of all quantum correlations. Instead, we position CE as a tractable, measurement-efficient summary statistic that preserves meaningful quantum structure while avoiding the impracticality of full state tomography. We explicitly clarify that matching distributions of CE is a necessary advancement over prior fixed-CE approaches, rather than a claim of sufficiency for capturing all quantum properties.

**Concerning the unintuitive difference between ideal and noisy QNN training results**, we conducted additional analysis and revised Sec. 6.6 to explain that the apparent performance reversal is within statistical variance across repeated runs. Both settings stabilize around approximately 82% accuracy, and the initial discrepancy does not reflect systematic noise benefits.

**On ansatz dependence and potential trial-and-error requirements**, we expanded Sec. 4.1 and Sec. 6.5 to clarify that QMill intentionally restricts itself to hardware-efficient ansatz families motivated by prior literature rather than performing open-ended architecture searches. We added concrete guidance showing how specific ansatz families align with particular CE distribution shapes. Users therefore do not need to perform costly tuning, and the framework remains immediately practical for current hardware.

**With respect to baselines and evaluation completeness**, we clarified in Sec. 6.1 that the absence of extensive external baseline comparisons stems from the novelty of the problem space itself. No prior methods target CE distribution matching. Our evaluation of the completeness of QMill is based on comprehensive multi-axis testing that examines distribution fidelity, noise robustness, sample diversity via SWAP tests, cross-hardware portability, ansatz sensitivity, stress testing on extreme distributions, and the downstream utility of the QML model. This multi-dimensional characterization remains fully meaningful even in the absence of pre-existing competing methods.

**To address generalization across hardware platforms and noise models**, we added new neutral-atom experiments and discussion in Sec. 4.4 and Sec. 5.2, including simulations under the Bloqade SDK using the QuEra Gemini computer. These experiments demonstrate comparable CE behavior across superconducting and neutral-atom backends, confirming that QMill’s shallow ansatzes generalize across hardware modalities.

**On concerns about QNN scale and system size**, we clarified in Sec. 5.2 that the three-qubit QNN appears only for the soil-moisture dataset, while most experiments utilize eleven-qubit circuits. Diversity validation through the SWAP test scales to $2n+1$ qubits and already reaches $21$ qubits for $n=10$, for example. Practical limits in these circuit sizes arise from classical simulation and the absence of large error-corrected devices, not from the QMill algorithm itself.

**We addressed questions surrounding amplitude encoding and PCA** in Sec. 5.2, clarifying that amplitude encoding is never used within the QMill generation pipeline. It appears only for classical dataset demonstrations. QMill’s true objective is to produce quantum-native datasets with both phase and entanglement structure, which amplitude encoding fundamentally cannot generate. PCA is applied solely to reduce classical feature dimensions so demonstrations remain feasible on current NISQ hardware and simulators. It is not required for quantum data generation.

---

> ### Author Response · Authors · 2025-11-30
> **Revisions and Clarifications Summary 2/2**
>
> **Regarding SWAP-test resource overhead**, Sec. 4.4 now explains that while SWAP testing doubles qubit count, it introduces only shallow three-qubit controlled operations that decompose efficiently into two-qubit gates. Current NISQ devices are fundamentally depth-limited rather than qubit-limited, making this overhead practical for both simulation and hardware execution.
>
> **We substantially strengthened the discussion of NZP scaling and surrogate CE estimation** in Appendix A.1. Because QMill uses deliberately shallow ansatz circuits rather than deep scrambling dynamics, the probability of the all-zero outcome remains bounded away from zero and does not vanish exponentially with qubit count. NZP estimation, therefore, reduces to Bernoulli mean estimation with shot complexity $O(\varepsilon^{-2})$, independent of the number of qubits. Importantly, NZP is used strictly as an ordering surrogate for CE and not for state reconstruction, ensuring polynomial cost throughout.
>
> Finally, we clarified in Sec. 6.1 that **the intentionally right-skewed CE stress-test distribution is physically unrealizable due to quantum reversibility**. Its inclusion reflects an adversarial stress test of QMill rather than an algorithmic limitation, and we now make this explicit.
>
> All notation issues and typographical errors identified by reviewers have been fully corrected.
>
> In summary, every reviewer's concern has been directly addressed through substantive clarification, new experiments, and improved theoretical framing. We are extremely thankful for the depth of reviewer engagement, which has meaningfully sharpened both the technical rigor and the narrative clarity of this work. We believe the revised manuscript now presents a carefully validated and robust contribution to the emerging field of quantum dataset generation, fully addressing all concerns raised during the review.

---

### Meta-Review · Area_Chair_GfNb · 2026-01-07

**Summary:**

This paper is a framework for generating Quantum data distributions. The main concern seems to be the utility of Synthetic Quantum Datasets for training Quantum Machine learning models does not seem to be clear. The more important discussion is to claim that this is a quantum dataset, the measure of concentratable entanglement(CE) seems to be a little restricted.

**Reviewer Concerns:**

Given the competitive forum like ICLR, it is important to demonstrate that there is significant scope in developing Quantum ML Algorithms with such synthetic dataset. Also a more comprehensive demonstration of CE as a measure of Quantum ness would be a great addition.

**Reviewer Scores:**

I guess no-one

---

### Decision · Program_Chairs · 2026-01-26

Reject